# SLY1 and Syntaxin 18 specify a distinct pathway for procollagen VII export from the endoplasmic reticulum

Cristina Nogueira[1,2†], Patrik Erlmann[1,2†], Julien Villeneuve[1,2], António JM Santos[1,2], Emma Martínez-Alonso[3], José Ángel Martínez-Menárguez[3], Vivek Malhotra[1,2,4*]

[1]Cell and Developmental Biology Program, Center for Genomic Regulation (CRG), Barcelona, Spain; [2]Universitat Pompeu Fabra (UPF), Barcelona, Spain; [3]Department of Cellular Biology and Histology, Faculty of Medicine, University of Murcia, Murcia, Spain; [4]Institució Catalana de Recerca i Estudis Avançats (ICREA), Barcelona, Spain

**Abstract** TANGO1 binds and exports Procollagen VII from the endoplasmic reticulum (ER). In this study, we report a connection between the cytoplasmic domain of TANGO1 and SLY1, a protein that is required for membrane fusion. Knockdown of SLY1 by siRNA arrested Procollagen VII in the ER without affecting the recruitment of COPII components, general protein secretion, and retrograde transport of the KDEL-containing protein BIP, and ERGIC53. SLY1 is known to interact with the ER-specific SNARE proteins Syntaxin 17 and 18, however only Syntaxin 18 was required for Procollagen VII export. Neither SLY1 nor Syntaxin 18 was required for the export of the equally bulky Procollagen I from the ER. Altogether, these findings reveal the sorting of bulky collagen family members by TANGO1 at the ER and highlight the existence of different export pathways for secretory cargoes one of which is mediated by the specific SNARE complex containing SLY1 and Syntaxin 18.

**\*For correspondence:** vivek. malhotra@crg.eu

[†]These authors contributed equally to this work

**Reviewing editor**: Suzanne R Pfeffer, Stanford University, United States

## Introduction

Collagens are the most abundant secretory proteins, comprising 25–30% of the human body dry weight (*Pataridis et al., 2008*). They are required for cell attachment, tissue organization and remodeling, and for the differentiation of chondrocytes to produce mineralized bones (*Gelse et al., 2003*; *Wilson et al., 2011*). There are at least 28 different kinds of collagens, composed of homo or hetero trimers of polypeptide chains coiled around each other to form a triple helix (*Shoulders and Raines, 2009*). These unbendable triple helices, which can be up to 450 nm long, as in the case of Collagen VII, are too big to fit into the conventional transport carriers of the secretory pathway that have been identified thus far (*Malhotra and Erlmann, 2011*). How are these bulky proteins exported from the ER?

While the debate on the trafficking of collagen-like molecules across the Golgi stack goes unabated, new data are beginning to unravel the mechanism by which collagens are exported from the ER. A protein called TANGO1 has been identified for its requirement in the export of Procollagen VII (PC VII) from the ER in tissue culture cells (*Saito et al., 2009*). The knockout of TANGO1 in mice results in the production of a pup that dies at birth due to defective bone mineralization. The cause is a block in the secretion of multiple collagens needed for the differentiation of chondrocytes (*Wilson et al., 2011*). TANGO1 is also required for collagen secretion in *Drosophila melanogaster* (*Pastor-Pareja and Xu, 2011*; *Lerner et al., 2013*). TANGO1 binds PC VII via its SH3 domain in the lumen of the ER (*Saito et al., 2009*). On the cytoplasmic side, TANGO1 binds cTAGE5 and both these proteins contain a proline rich domain that interacts with the COPII components SEC23/24 (*Saito et al., 2009*,

**eLife digest** Collagens are long proteins that join individual cells together to build tissues and organs. They also provide strength and elasticity to bones, tendons, and blood vessels. Like many other proteins, collagens are produced inside cells: they are folded in a compartment called the endoplasmic reticulum, and then packaged and transported to another compartment called the Golgi. Collagens are then directed from the Golgi to their final destination, which is typically the outside of the cell.

Small proteins travel from the endoplasmic reticulum to the Golgi inside packages called vesicles. However it is not clear how large proteins like collagens are transported between these two compartments. It is known that a protein called TANGO1 is needed to direct a collagen called Procollagen VII to the outside of the cells. TANGO1 binds to Procollagen VII, and it is thought that TANGO1 delays the release of Procollagen VII from the endoplasmic reticulum, so that the vesicle can grow to a size that is able to accommodate such a bulky cargo.

Nogueira, Erlmann et al. have now discovered that TANGO1 binds to another protein called SLY1, and that this protein must also be present if Procollagen VII is to be exported from the endoplasmic reticulum. In contrast, the transport of a different type of collagen—Collagen I—does not require TANGO1 or SLY1.

SLY1 helps to fuse the membranes that enclose the structures involved in protein trafficking—such as the endoplasmic reticulum, the Golgi, and the vesicles—and this allows the cargoes of vesicles to pass from one compartment to another. Nogueira, Erlmann et al. also found that a second protein (called Syntaxin 18) is also required for the export of Procollagen VII.

Nogueira, Erlmann et al. propose that collagen VII export involves TANGO1 delaying the release of collagen from the endoplasmic reticulum so that SLY1 and Syntaxin 18 can fuse other cellular membranes to the growing transport vesicle. Following this work, the next challenge is to uncover how different types of collagens are separated from each other, and identify which specific vesicles are involved in their export.

*2011*). We have proposed that binding of PC VII to TANGO1 in the lumen promotes the binding of TANGO1's proline rich domain to SEC23/24. This retards the recruitment of SEC13/31 to SEC23/24 and thus delays the events leading to the biogenesis of the COPII vesicle (*Malhotra and Erlmann, 2011*). Upon growth to a size that is large enough to encapsulate PC VII, TANGO1 dissociates from both PC VII and SEC23/24. The binding of SEC13/31 to SEC23/24 completes the assembly of COPII components on a patch of the ER. These events then lead to the export of PC VII, presumably in a mega carrier from the ER (*Saito et al., 2009, 2011*). Ubiquitination of SEC31 by the CUL3-KLHL12 ligase complex has been reported to control the exit of Procollagen I (PC I) from the ER (*Jin et al., 2012*; *Malhotra, 2012*). Sedlin is reported to help in the export of PC I and II from the ER by regulating the cycling of SAR1 activation state that is essential for COPII assembly at the ER (*Venditti et al., 2012*). TANGO1 is not required for PC I export from the ER, and it is not known whether PC II export is TANGO1 dependent. Together these data indicate that COPII components are required for the export of procollagens from the ER, however, they also suggest the possibility that not all procollagens exit the ER by the same mechanism.

We now show the involvement of SLY1 (or SCFD1) in specific ER export events. SLY1 is a member of the STXBP/unc-18/SEC1 family of proteins that regulate the assembly or the activity of SNAREs in membrane fusion events (*Carr and Rizo, 2010*). The yeast ortholog *SLY1*, an essential gene, has been described as a single copy suppressor of the *YPT1* deletion (*Dascher et al., 1991*) and implicated in forward and retrograde trafficking (*Ossig et al., 1991*; *Li et al., 2005*). In contrast with its essential roles in yeast, a temperature sensitive mutant of Sly1 in zebra fish is not lethal on the cellular level but rather creates developmental defects in embryonic stages (*Nechiporuk et al., 2003*). In mammals, SLY1 has been reported to function in conjunction with Syntaxin 5 (STX5) in the ER to Golgi transport and might also function in the assembly of pre-Golgi intermediates (*Rowe et al., 1998*) together with Syntaxin 18 (STX18) (*Yamaguchi et al., 2002*) and Syntaxin 17 (STX17) (*Steegmaier et al., 2000*). SLY1 has been shown to interact with the COG4 complex and suggested to play a role in intra Golgi and retrograde transport (*Laufman et al., 2009*). It is important to note that in mammalian cells, these

proposed roles of SLY1 in traffic between ER and Golgi membranes are based entirely on the use of artificial temperature sensitive mutant protein Vesicular Stomatitis Virus (VSV)-Glycoprotein (G) protein and the artificial cargo signal sequence (ss)-Green Fluorescent Protein (GFP). The role of SLY1 in the trafficking of endogenous cargoes and its potential mechanism of action is therefore a matter of debate.

We describe in this study, our data that reveal the existence of different export routes for secretory cargoes from the ER: of specific interest is the finding that SLY1 and the ER specific t-SNARE STX18 are necessary for the export of PC VII but not of the equally bulky PC I from the ER.

## Results

### SLY1 is cross-linked to the ER exit sites specific TANGO1 and localizes to ER exit sites

To search for proteins that interact with cytoplasmically oriented portions of TANGO1, we expressed a Myc-His tagged version of a truncated form of TANGO1 (TANGO1ct) that lacks the luminal domain in HeLa cells. After crosslinking with membrane permeable DSP and lysis, proteins were recovered on a Nickel agarose column and analyzed by mass spectrometry. Of interest was the finding of SLY1 in the pool of proteins cross-linked to TANGO1.

To further ascertain the mass spectrometry data, we immunoprecipitated Myc-His-tagged TANGO1ct from transfected and crosslinked HeLa cells as described above and western blotted the bound material with anti-Myc and SLY1 antibodies, respectively. Our data show the presence of SLY1 in the TANGO1 immunoprecipitate (*Figure 1A*). SLY1 is a cytoplasmic protein but our findings suggest that it interacts with the ER exit sites anchored TANGO1, so is there a pool of SLY1 associated with ER exit sites where TANGO1 resides? We have previously documented the use of a human cell line called RDEB/FB/C7 that stably expresses and secretes Collagen VII (*Chen et al., 2002*; *Saito et al., 2009*). We used lentivirus to express SLY1-GFP in RDEB/FB/C7 cells. In fixed cells, the GFP signal is localized throughout the cytoplasm with a slight accumulation in the perinuclear region. After prepermeabilization of the cells with saponin and extensive washes to remove the cytoplasmic pool of proteins before fixation, a pool of SLY1-GFP was found to colocalize with the ER exit sites specific COPII component SEC31 (*Figure 1B*). Arrowheads mark single exit sites that label for SEC31 and SLY1. In addition, a perinuclear SLY1-containing and SEC31-lacking pool was observed. This perinuclear pool of SLY1 partially overlaps with GM130, a marker of the early Golgi cisternae (*Figure 1B*). Our data show that although most of SLY1 was Golgi localized, we detected a small pool that co-localized with Sec31. At present, we do not know whether recruitment of SLY1 to the ER exit site is by direct binding to TANGO1 or indirectly through other proteins on the cytoplasmic side.

### SLY1 is required for procollagen VII secretion

The identification of SLY1 in a pool of proteins that are cross linked to TANGO1 prompted us to test its involvement in PC VII export from the ER. In addition to the RDEB/FB/C7 cell line in which Collagen VII was exogenously introduced, we tested the immortalized human esophageal cell line Het1a to monitor the involvement of SLY1 in the export of endogenous PC VII. RDEB/FB/C7 and Het1a cells were transfected with siRNA oligos specific for SLY1, TANGO1, or a scrambled oligo as a control. RDEB/FB/C7 cells were transfected twice to increase knockdown efficiency. The levels of SLY1 and TANGO1 after knockdown were monitored by western blotting and revealed an approximately 90% reduction compared with control cells. Knockdown of TANGO1 did not significantly alter SLY1 protein levels and vice versa (*Figure 2A*). 48 hr after the last siRNA transfection, the cells were washed and then incubated with fresh medium containing ascorbic acid (2 µg/ml) for 20 hr. The medium and the cell lysates were western blotted with an anti-Collagen VII antibody. Knockdown of SLY1 inhibited PC VII secretion (*Figure 2B*). Quantitation of the ratio of the secreted to intracellular PC VII revealed a 75% reduction upon SLY1 knockdown compared with control cells. PC VII secretion was 90% reduced upon TANGO1 knockdown, compared with control cells (*Figure 2C*).

We then transfected RDEB/FB/C7 or Het1a cells with SLY1 siRNA, TANGO1 siRNA, or a control siRNA, as described above, treated the cells with ascorbic acid and after 72 hr the cells were fixed and incubated with antibodies against Collagen VII, the ER-resident chaperone Hsp47 and GM130. In control cells, we did not detect intracellular PC VII because it is rapidly exported from the ER and secreted by the cells. In SLY1 knockdown cells, on the other hand, PC VII accumulated in large structures that

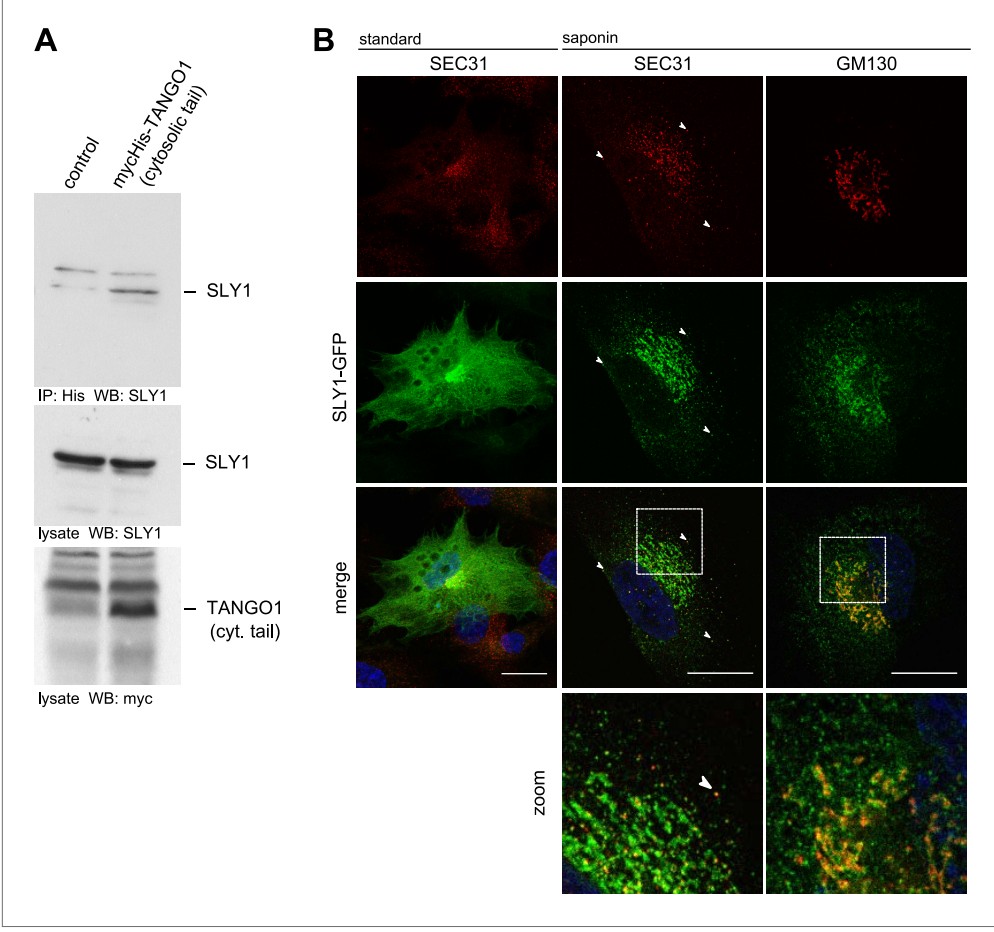

**Figure 1**. SLY1 immunoprecipitates with TANGO1 and localizes to ER exit sites. (**A**) Lysates from Myc-His-TANGO1 (cytoplasmic tail) transfected or control HeLa cell lysates were incubated with Nickel beads, bound proteins (75%) and total lysates (10%) were separated by SDS-PAGE, endogenous SLY1 and Myc-His-TANGO1 (cytoplasmic tail) was detected by Western Blot using SLY1 and Myc antibodies. (**B**) RDEB/FB/C7 cells stably expressing SLY1-GFP (green) were fixed with 4% PFA and permeabilized with Triton-X100 or pre-extracted with saponin, washed and then fixed with 4% PFA prior to permeabilization with Triton-X100. Cells were labeled with an anti-SEC31 or an anti-GM130 antibody (red) and DAPI (blue, Scale bar: 20 μm); arrowheads indicate colocalization of SLY1-GFP and SEC31.

also contained the collagen-specific protein chaperone Hsp47 (*Saga et al., 1987*). Similar large patches of PC VII are also evident in RDEB/FB/C7 cells depleted of TANGO1, which is well known for its role in the export of PC VII from the ER (*Saito et al., 2009*, *2011*) (*Figure 2D*, upper panel). We did not observe accumulation of PC VII in the Golgi membranes upon knockdown of either SLY1 or TANGO1 (*Figure 2D*, lower panel). The very low levels of Collagen VII expressed in Het1a cells are difficult to detect by the standard western blotting approaches, however, the accumulation of PC VII in the ER and colocalization with Hsp47 by fluorescence microscopy is also evident in SLY1- and TANGO1-depleted Het1a cells (*Figure 2E*). Based on these data, we suggest that SLY1 knockdown arrests PC VII in the ER where it accumulates in large patches.

To further ascertain the site of PC VII localization in SLY1 knockdown cells, we visualized PC VII by immunoelectron microscopy. In control cells, gold-conjugated secondary antibodies used to visualize anti-Collagen VII antibody were found distributed throughout the ER. In SLY1 knockdown cells, however, PC VII was observed in large patches within dilated ER (*Figure 3*) but smaller vesicles that could correspond to intermediates en route were not visible. These patches, we suggest, correspond to the patches of PC VII observed in the fluorescence micrographs of SLY1 knockdown cells (*Figure 2D*). Similar patches of PC VII were also evident upon knockdown of TANGO1 (*Figure 3*).

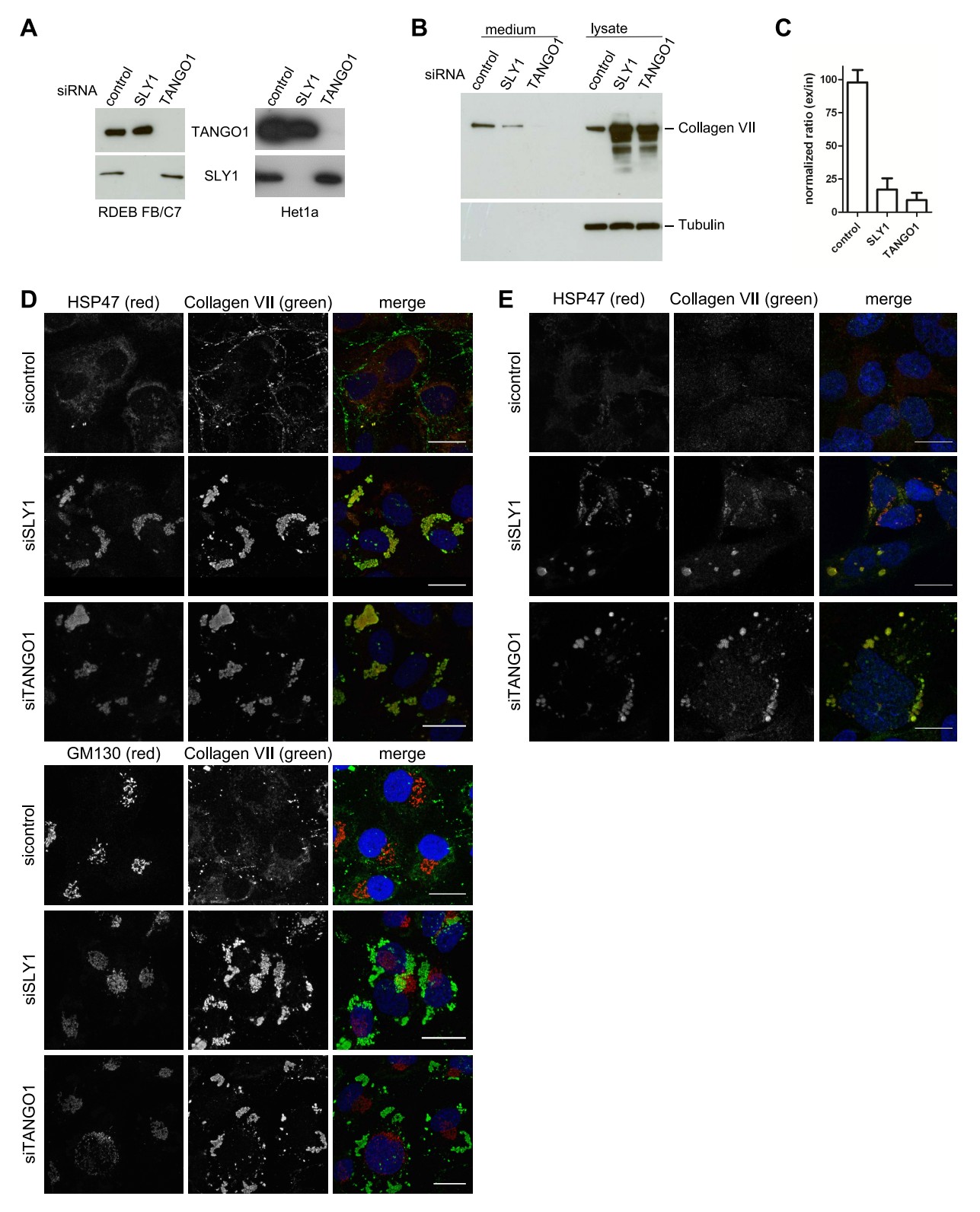

**Figure 2**. SLY1 knockdown by siRNA inhibits Procollagen VII secretion. RDEB/FB/C7 cells were transfected with siRNAs directed against SLY1, TANGO1, or a scrambled siRNA. Het1a cells were transfected with the same set of siRNAs. (**A**) Knockdown efficiency was tested after 72 hr by western blotting cell lysates with anti-TANGO1 or anti-SLY1 antibodies. (**B**) PC VII secretion was measured by western blotting RDEB/FB/C7 cell

*Figure 2. Continued on next page*

*Figure 2. Continued*
lysates and supernatants collected for 20 hr in the presence of ascorbic acid with an anti-Collagen VII antibody. Equal protein loading and cell lysis were controlled by blotting with an anti-Tubulin antibody. (C) In three independent experiments, intensities of the PC VII signal in the lysate and the supernatant was recorded by densitometry. The ratio of external vs internal Collagen VII was normalized to quantify secretion in control cells as 100%; Error bars: standard error of the mean (SEM). siRNA-treated RDEB/FB/C7 (D) or Het1a (E) cells were seeded on coverslips and 20 hr after addition of ascorbic acid, cells were fixed and visualized with the indicated antibodies and DAPI (blue) by fluorescence microscopy (scale bars: 20 μm).

## SLY1 knockdown does not affect the overall levels of COPI and COPII components

We then tested whether SLY1 knockdown affected the stability of proteins involved in the transport of cargoes between the ER and the Golgi. For this, we monitored the levels of COPI and COPII components in the lysates of SLY1, TANGO1 knockdown or control siRNA-treated RDEB/FB/C7 cells. The level of SAR1, the initiator of the COPII vesicle budding process, did not change upon SLY1 or TANGO1 knockdown. Also unchanged were the levels of the other COPII components that are required for the export of secretory cargo, including components of the extracellular matrix, from the ER, specifically SEC23A (*Boyadjiev et al., 2006*) and SEC31 (*Jin et al., 2012*). Because of the possible involvement of SLY1 in the retrograde trafficking pathway, we also tested components of the COPI coat: β-COP and ε-COP, and their levels were also unchanged (*Figure 4*).

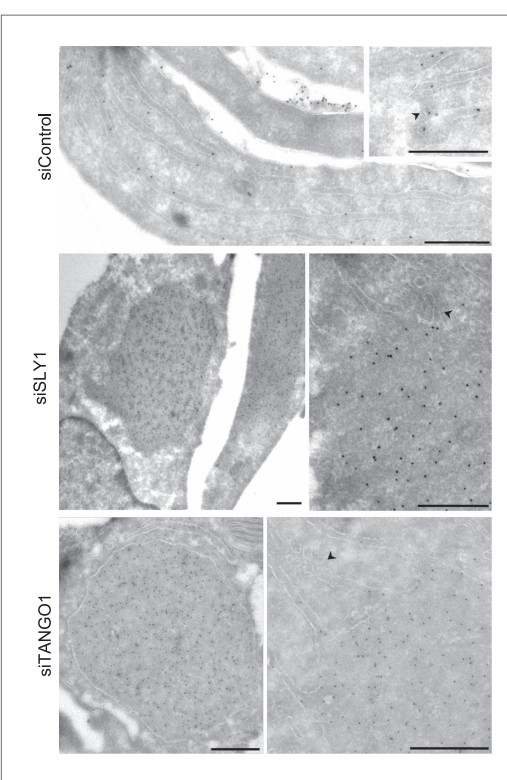

**Figure 3**. Immunoelectron microscopy localizes intracellular Procollagen VII in SLY1 knockdown cells to the ER. Control cells and cells depleted for SLY1 or TANGO1 were fixed for immunoelectron microscopy. Ultrathin cryosections were labeled with a polyclonal rabbit antibody against Collagen VII followed by 10 nm protein A-gold complex. Arrowheads mark ER exit sites. Bars, 250 nm.

## SLY1 is not required for general protein recycling to the ER

We tested the involvement of SLY1 in retrograde trafficking by monitoring the localization of a protein called GRP78/BIP that contains a KDEL sequence and cycles between the ER and the early Golgi cisternae. A block in retrograde transport leads to an accumulation of GRP78/BIP in the Golgi apparatus and its subsequent secretion from the cells (*Yamamoto et al., 2003*). GRP78/BIP was localized to the ER in SLY1 knockdown cells and control cells (*Figure 5A*). Moreover, GRP78/BIP was not detected in the medium collected from control, SLY1- or TANGO1-depleted cells (*Figure 5B*). To further quantitate the involvement of SLY1 in transport between the Golgi apparatus and the ER, we used a procedure to trap cycling proteins in the ER as described previously (*Pecot and Malhotra, 2006*). HeLa cells treated with control siRNA or cells depleted of TANGO1 or SLY1 were transfected with the following constructs: one encoding ERGIC53, a protein that cycles between the early Golgi/ERGIC compartment and the ER, fused to FKPB and GFP, and another encoding the invariant chain of the major histocompatibility complex class II receptor (Ii) fused to FRAP-HA. In HeLa cells, the Ii chain lacking its MHC counterpart is retained in the ER (*Lotteau et al., 1990*). Rapamycin is known to bind FRAP and this dimer tightly binds FKBP. This procedure can therefore be used to test if FRAP- and FKBP-containing proteins reside in close proximity and thus are accessible to each other. We observed the cells by fluorescence

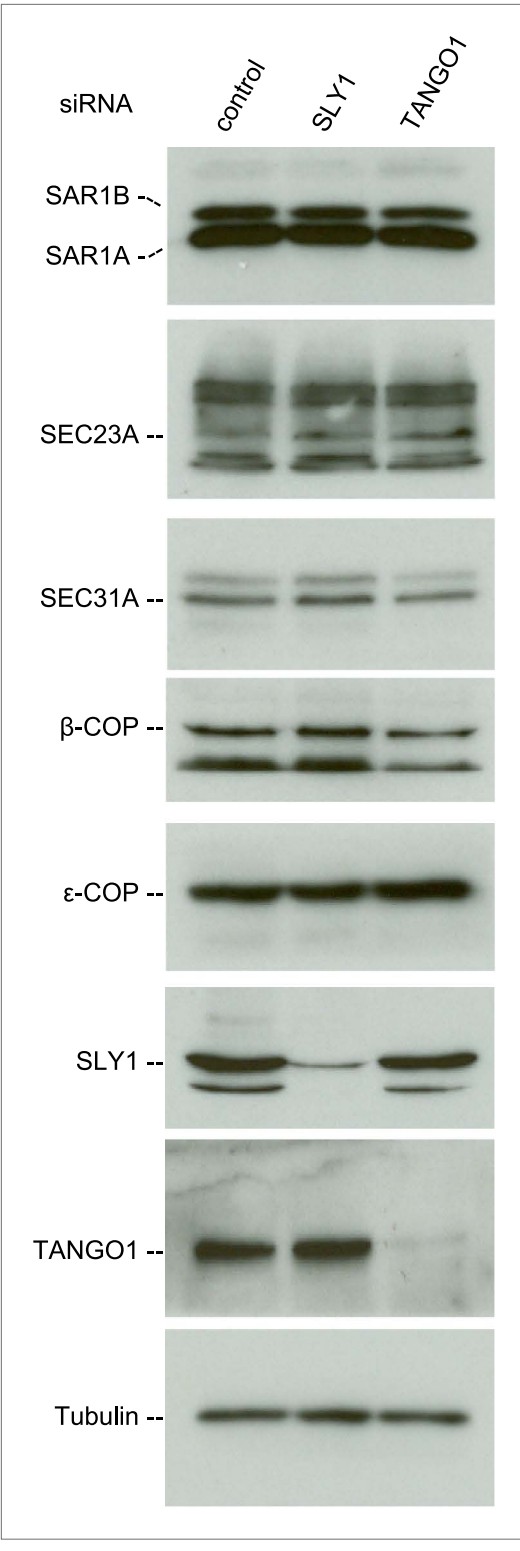

**Figure 4**. SLY1 knockdown does not affect the levels of COPI and COPII components. Cell lysates from RDEB/FB/C7 cells transfected with siRNAs oligos against SLY1, TANGO1, or a scrambled siRNA were western blotted with antibodies to the indicated COPI and COPII components.

microscopy at steady state and 15, 30, and 60 min after the addition of Rapamycin or in control cells without the addition of Rapamycin. Colocalization of the two constructs was assessed by calculating the Pearson's colocalization coefficient. Knockdown of TANGO1 or SLY1 did not affect the kinetics of ERGIC53 redistribution to the ER (*Figure 5C*). We therefore conclude that under our experimental conditions, knockdown of SLY1 or TANGO1 does not affect retrograde trafficking.

Retention of PC VII in the ER, upon knockdown of SLY1 or TANGO1, is therefore not because of a defect in cycling of a component that is specifically required for the constitutive trafficking between the ER and the early Golgi cisternae.

## SLY1 knockdown does not affect secretion of bulk-endogenous-cargoes

Is bulk protein secretion affected by SLY1 knockdown? RDEB/FB/C7 cells were transfected with SLY1, TANGO1 or a scrambled siRNA oligo, and after 72 hr, the cells were washed, cultured in methionine and cysteine free medium for 1 hr, pulsed with $^{35}$S-methionine for 20 min in methionine free medium, washed and then incubated in complete medium. Another control was included in which scrambled siRNA oligo transfected cells were incubated with Brefeldin A (BFA) during the chase period. The medium was collected after 2 hr and then analyzed by SDS-PAGE/autoradiography to identify the secreted polypeptides. Treatment with BFA, as expected, inhibited the secretion of newly synthesized proteins. Surprisingly, SLY1 knockdown did not have any appreciable effect on the secretion of newly synthesized proteins (*Figure 6A*). This approach does not reveal the specific polypeptides secreted in a SLY1-dependent manner, but shows that the overall pattern of polypeptides detected by SDS-PAGE is not affected by SLY1 knockdown.

It has been shown that SLY1 depletion inhibits the trafficking of widely used model cargoes such as the temperature sensitive VSVG (*Dascher and Balch, 1996*) and signal sequence GFP (*Gordon et al., 2010*). We tested the effect of SLY1 and TANGO1 depletion on the secretion of signal sequence—Horseradish peroxidase (ss-HRP). HeLa cells stably expressing ss-HRP were transfected with siRNAs targeting either SLY1 or TANGO1 or a scrambled siRNA as a control. After 72 hr, cells were washed and ssHRP was accumulated in the ER for 2 hr by incubating cells at 15°C. New protein synthesis was then blocked by Cycloheximide (CHX) treatment and the cells were incubated for the indicated time

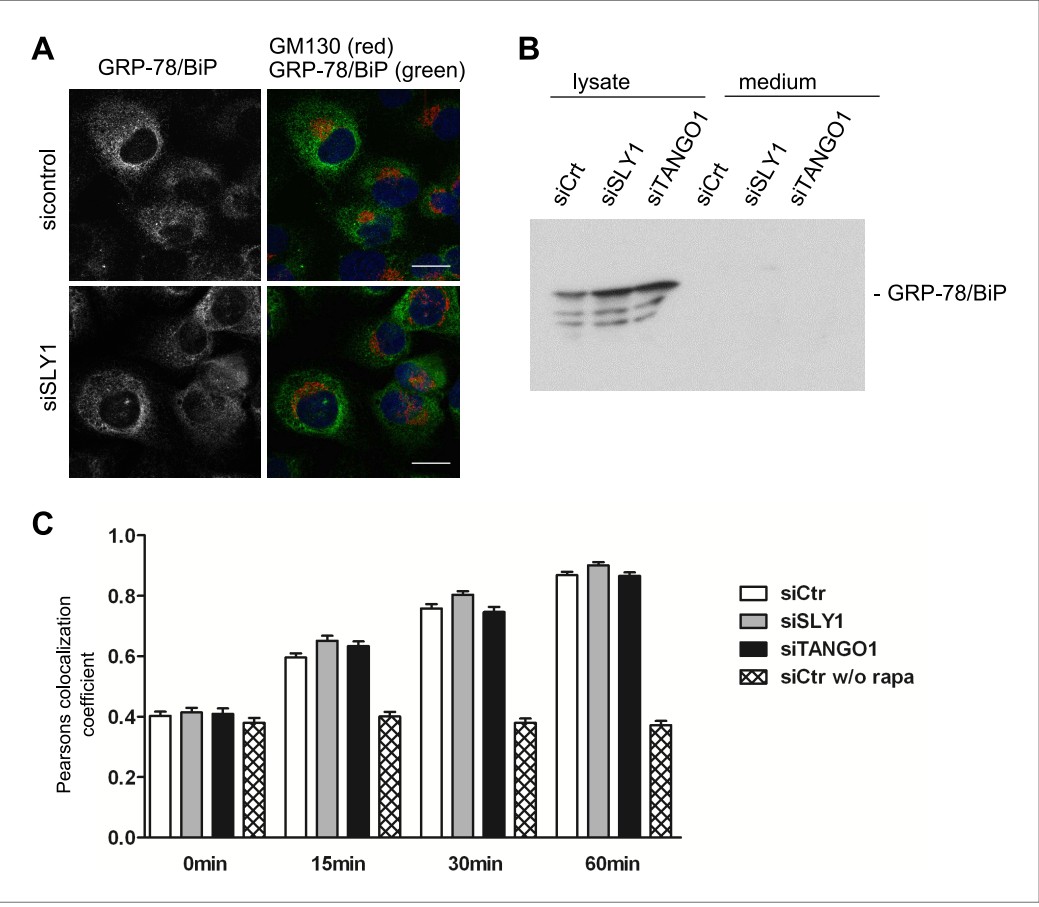

**Figure 5**. SLY1 knockdown does not affect retrograde trafficking. (**A**) RDEB/FB/C7 cells depleted of SLY1 or control cells were visualized using an anti-GRP78/BIP, an anti-GM130 antibody and DAPI by fluorescence microscopy (scale bars: 20 μm). (**B**) Lysates and 20 hr medium of SLY1, TANGO1 or control knockdown RDEB/FB/C7 cells were analyzed by SDS-PAGE and western blotted with an anti-GRP78/BIP antibody. (**C**) In HeLa cells transfected with the indicated siRNAs, the colocalization of FKBP-ERGIC53-GFP and the ER localized Ii-FRAP-HA was determined at each indicated time points after addition of Rapamycin. Trapping, and thus colocalization, was measured by calculating the Pearsons colocalization coefficient between the GFP signal and the Alexa594-stained Ii-FRAP-HA. The average values of at least 30 cells analyzed per experiment for each condition are shown; Error bars: SEM.

at 37°C to restart export and trafficking of secretory cargo from the ER. The HRP activity in the lysate and the supernatant was measured as described previously (*Bard et al., 2006*). The ratio of secreted to internal HRP shows that depletion of SLY1 efficiently blocked secretion ss-HRP and was comparable to known secretion inhibitory drug BFA. TANGO1 depletion reduced the secretion of ss-HRP to 50% of that from control cells (*Figure 6B*).

## SLY1 knockdown does not affect procollagen I export from the ER

The data presented thus far reveal the requirement of SLY1 in PC VII export, the export of exogenously expressed secretory cargo ss-HRP, but not the bulk of endogenous secretory cargoes. This is similar to the known effects of TANGO1 knockdown on the selective export of secretory cargo from the ER. Interestingly, TANGO1 knockdown by siRNA does not affect the secretion of the bulky PC I (*Saito et al., 2009*, *2011*). Is SLY1 required for the trafficking of PC I? RDEB/FB/C7 cells were transfected with siRNA oligos specific for SLY1, TANGO1 or scrambled siRNA as a control and treated with ascorbic acid, as described above. 20 hr after the last change of medium, the cell lysates and the medium were western blotted with anti-Collagen I antibody. Our results reveal that the secretion of collagen I is not blocked by knockdown of SLY1 or TANGO1 (*Figure 6C*). To further ascertain that SLY1 has no role in PC I export, we transfected Saos 2 cells with siRNA oligos specific for SLY1, TANGO1 or

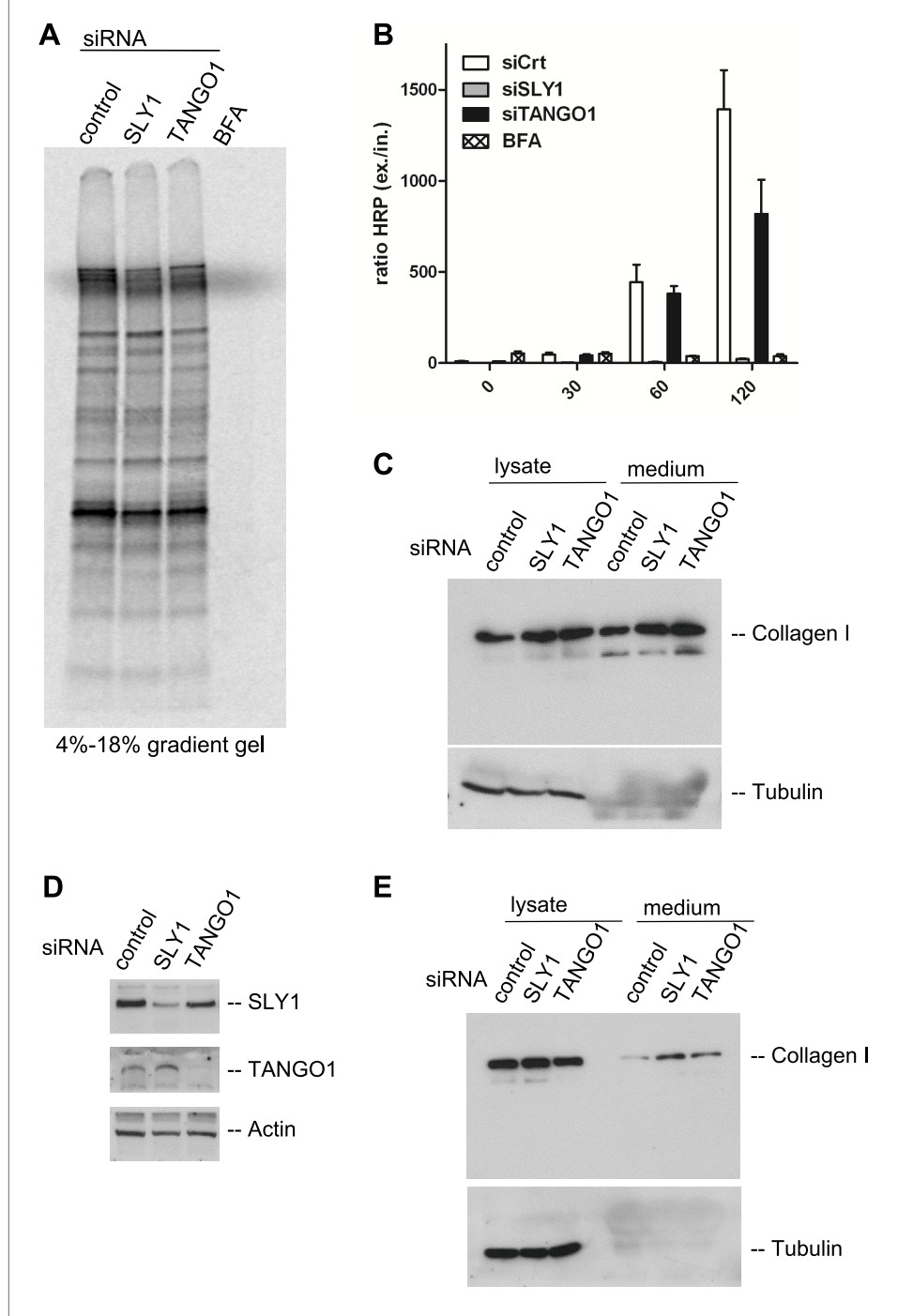

**Figure 6**. SLY1 knockdown does not block the export of endogenous secretory cargo. (**A**) RDEB/FB/C7 cells depleted for SLY1 or TANGO1 or control cells were pulsed with $^{35}$S-methionine for 20 min and chased for 2 hr in complete medium that included 5 mg/ml BFA where indicated. The medium from the cells was collected and analyzed by SDS-PAGE and autoradiography. (**B**) HeLa ss-HRP cells were transfected with control siRNA, SLY1 or TANGO1 specific siRNA oligos or treated with BFA as a positive control for total block in secretion. At the indicated time points, the medium and cell lysates were harvested to measure HRP activity. The graph shows the ratio of secreted to intracellular ss-HRP activity. Average values of three independent experiments are shown; Error: SEM. (**C**) RDEB/FB/C7 cells were transfected with control siRNA, SLY1 or TANGO1 specific siRNA oligos. Collagen I secretion was measured by western blotting of RDEB/FB/C7 cell lysates and supernatants collected for 20 hr in the presence of ascorbic acid with an anti-Collagen I antibody. The samples were western blotted with an anti-Tubulin
*Figure 6. Continued on next page*

*Figure 6. Continued*

antibody to monitor loading control and cell lysis. (**D**) Saos2 cells were transfected with control, SLY1 or TANGO1 siRNAs. Knockdown efficiency was tested after 72 hr by western blotting cell lysates with anti-TANGO1 or anti-SLY1 antibodies. Actin was used as a loading control. (**E**) Collagen I secretion was measured by western blotting Saos2 cell lysates and supernatants collected for 20 hr in the presence of ascorbic acid with an anti-Collagen I antibody. The samples were also western blotted with an anti-Tubulin antibody to monitor loading control and cell lysis.

a scrambled siRNA and after 48 hr Saos 2 cells were washed, and then incubated with fresh medium containing ascorbic acid for 20 hr. The knockdown efficiency was determined by western blotting the cell lysates and revealed a reduction greater than 75% in the levels of SLY1 and TANGO1 (*Figure 6D*). The intracellular and secreted PC I was measured by western blotting the cell lysate and the medium, respectively. As in RDEB/FB/C7 cells, knockdown of SLY1 or TANGO1 in Saos 2 cells did not affect the secretion of PC I (*Figure 6E*). In fact, in both cells lines, we detected a marginal increase in the secretion of Collagen I that is accompanied by an increase in the internal levels of Collagen I in the knocked down cells. This suggests a potential effect on the transcriptional regulation of Collagen I expression rather than a defect in its trafficking.

## SAR1 activity is necessary for procollagen VII export

The assembly of the COPII coat is initiated by the activation of the small GTPase SAR1 by the ER-resident protein SEC12. Active SAR1 recruits the inner coat proteins SEC23/24. The assembly of the outer coat proteins SEC13/31 completes the biogenesis of COPII vesicles by inactivation of SAR1 (*Zanetti et al., 2012*). TANGO1 links PC VII to the inner COPII coat by interacting with the SEC23/24 complex (*Saito et al., 2009*). To further ascertain the involvement of COPII coat assembly at the ER for PC VII export, we depleted either of the SAR1 isoforms A and B or both in RDEB/FB/C7 cells using an siRNA mixture described earlier (*Cutrona et al., 2013*). Probing of cell lysates with anti SAR1 antibody revealed a doublet of bands at the predicted size of SAR1. As one can observe in the single knockdowns these bands correspond to the two SAR1 isoforms, the upper, weaker band being SAR1 B and the lower, stronger SAR1 A. Single knockdowns result in an almost complete depletion of the individual isoforms, with double knockdown the efficiency is approximately 75% for SAR1 A while SAR1 B is still fully depleted (*Figure 7A*). The same procedure of SAR1 knockdown was repeated and 48 hr after transfection with the specific siRNA oligos, the cells were washed, and then incubated with fresh medium containing ascorbic acid for 20 hr. The medium and the cell lysates were western blotted with an anti-Collagen VII antibody. Knockdown of SAR1 A alone had marginal effect on the secretion of PC VII, whereas the knockdown of SAR1 B was largely ineffective. However, knockdown of both SAR1 A and B led to an almost total block in PC VII secretion, which accumulated in the cells (*Figure 7B*). To visualize the intracellular site of PC VII accumulation, we transfected RDEB/FB/C7 cells to deplete both SAR1 A and B as described above, and 24 hr later ascorbic acid was added to the culture medium. 48 hr later, the cells were fixed and stained with antibodies against Collagen VII and SEC31, respectively. In cells depleted of SAR1 A and B, PC VII accumulated in large structures (*Figure 7C*) similar to the patches of PC VII seen in RDEB/FB/C7 cells depleted of TANGO1 or SLY1 (*Figure 2D*). As expected the COPII component SEC31 localized to punctate structures resembling ER exit sites in control cells, but not in cells depleted of SAR1 A and B. Here, SEC31 localized mostly in the cytosol and almost no SEC31 positive punctae were visible (*Figure 7C*).

## SLY1 knockdown does not affect Sec13/Sec31 localization to the ER exit sites

Recruitment of SEC13/31 to SEC23/24 is the ultimate known event in the assembly of COPII coats that leads to membrane fission and the separation of COPII vesicles from the ER. Is SLY1 activity required before or after the COPII recruitment to the ER exit sites? TANGO1, as shown previously, connects PC VII to the COPII subunits through interaction with SEC23/24. The question we asked was whether the location of COPII components and PC VII in cells that are depleted of SLY1 is different compared with control cells that are exporting Collagen VII. Since PC VII is rapidly exported from the ER after its synthesis, our positive control, to test this hypothesis, was cells transfected with scrambled siRNA oligos

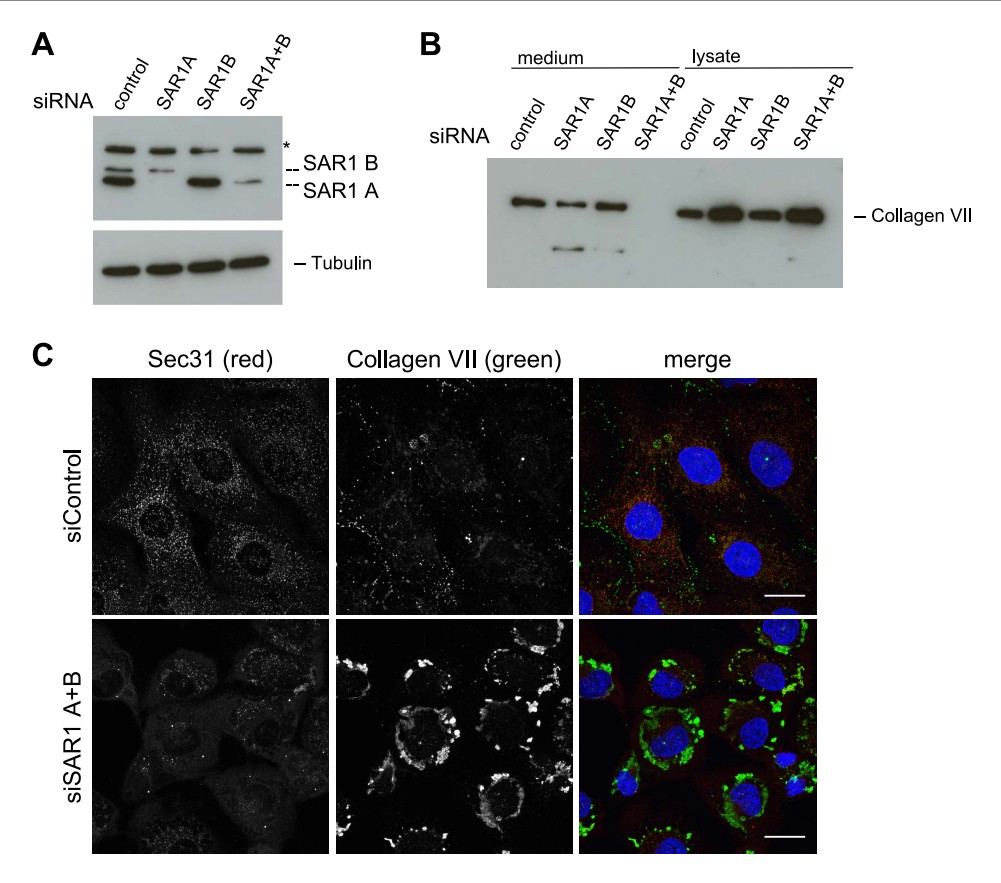

**Figure 7**. SAR1 A and B knockdown blocks Collagen VII secretion. RDEB/FB/C7 cells were transfected with siRNAs directed against SAR1A, SAR1B, both or a scrambled control siRNA. (**A**) Knockdown efficiency was determined after 72 hr by western blotting cell lysates with an anti-SAR1 antibody. Tubulin was used as a loading control. *-unspecific band. (**B**) Collagen VII secretion was measured by western blotting RDEB/FB/C7 cell lysates and supernatants collected for 20 hr in the presence of ascorbic acid using an anti-Collagen VII antibody. (**C**) RDEB/FB/C7 cells treated with ascorbic acid were seeded on coverslips and 72 hr after transfection with control or SAR1 A and B siRNA, the cells were fixed and visualized by fluorescence microscopy with the indicated antibodies and DAPI (blue, scale bars: 20 μm).

for 72 hr and then kept for 3 hr at 15°C to accumulate secretory cargo in the ER and then shifted to 37°C for 30 min to restart cargo export from the ER. These cells were visualized by fluorescence microscopy to monitor the location of PC VII and either the COPII component SEC31 or TANGO1. In the control cells, PC VII accumulated in the ER in large patches similar to that observed in SLY1 or TANGO1 knockdown cells. The majority of SEC31 and TANGO1 were localized to specific sites on the ER that contained large patches of PC VII (***Figure 8A***).

We then compared the location of SEC31 and TANGO1 in RDEB/FB/C7 cells that had been transfected to knockdown specifically SAR1A and B, TANGO1, or SLY1 by fluorescence microscopy with anti-Collagen VII and SEC31 or TANGO1 antibodies (***Figure 8B–D***). In SAR1 (A and B) knockdown cells, as expected, SEC31 was distributed mostly in the cytoplasm and not associated with the ER exit sites. Interestingly TANGO1 was still present at the collagen-enriched patches, suggesting that the TANGO1–Collagen VII binding is independent of TANGO1–SEC23/24 interaction (***Figure 8B***). In TANGO1 knockdown cells, the SEC31 positive ER exit sites appeared fewer in number on the collagen patches and more random in localization (***Figure 8C***). Interestingly, in SLY1-depleted cells, the number and localization of SEC31 sites resembled that of control cells (***Figure 8D***). SLY1 knockdown did not affect the location of TANGO1 at the ER exit sites. These findings suggest that knockdown of SLY1 does not interfere with the capture of PC VII by TANGO1 nor with the assembly of a complete COPII coat.

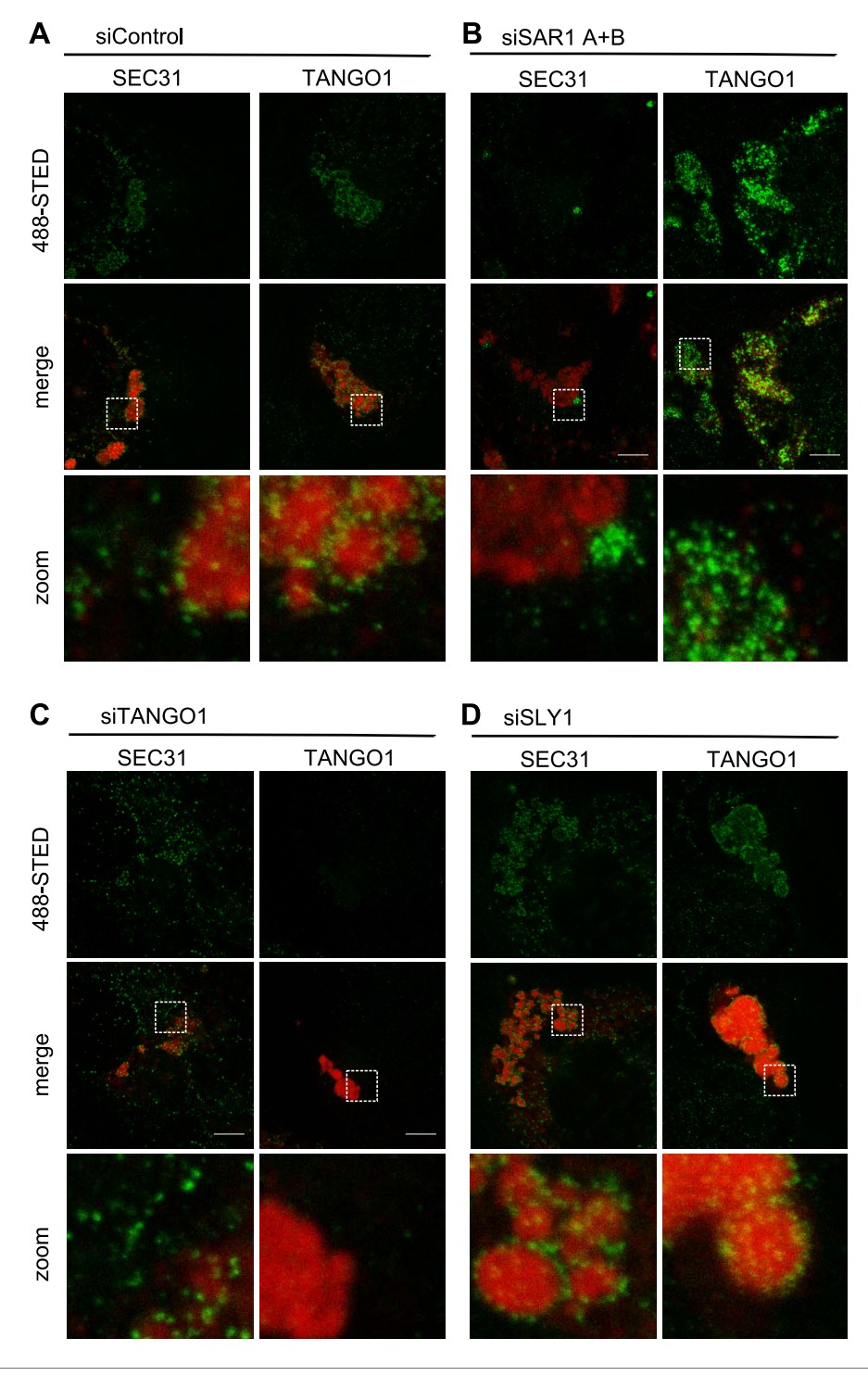

**Figure 8**. SLY1 is required for post coat assembly events in PC VII export. RDEB/FB/C7 cells were transfected with control siRNA (**A**) or siRNAs directed against SAR1 A+B (**B**), TANGO1 (**C**) or SLY1 (**D**). Cells were incubated in medium containing ascorbic acid for 72 hr. In control cells (**A**) protein export from the ER was arrested for 3 hr at 15°C and cells were fixed after a 30 min release to 37°C. All other samples were fixed without a 15°C temperature block (**B**–**D**). The ER exit sites were visualized using either anti-SEC31 or anti-TANGO1 antibodies (green) in STED mode. PC VII patches were visualized using either anti-Collagen VII (in combination with anti-SEC31) or the colocalizing HSP47 (in combination with anti-TANGO1) antibodies (red) in normal confocal mode. Edge length of the zoom boxes is approximately 5 µm.

# Syntaxin 18 and Syntaxin 5 are required for the export of procollagen VII from the ER

A large number of SNAREs have been localized and reported to traffic cargoes between the ER and the Golgi (*Table 1*). SLY1 interacts with several of these ER to Golgi SNARE proteins (*Peng and Gallwitz, 2004*). To further elucidate the function of SNAREs in the export of PC VII, we tested SLY1 interacting SNAREs, specifically the ER localized STX18 and STX17 as well as the ER and Golgi membrane associated STX5 (*Rowe et al., 1998*; *Steegmaier et al., 2000*; *Yamaguchi et al., 2002*). RDEB/FB/C7 or Saos2 cells were transfected with siRNA oligos for the respective t-SNAREs. The knockdown efficiency of the cognate mRNAs was assessed by rtPCR (*Figure 9A*). The medium from RDEB/FB/C7 cells treated with siRNAs for STX5, 17 and 18 was collected as previously described and western blotted for the presence of PC VII. We observe that both knockdown of STX5 and 18 reduced the amount of PC VII being secreted. STX17 had only marginal effect on the secretion of PC VII (*Figure 9B,C*). The secretion of Collagen I to the media was assessed in Saos2 cells. We observed a reduction of secreted protein, when cells were treated with siRNAs for STX5. Neither depletion of STX17 nor depletion of STX18 affected the levels of secreted Collagen I (*Figure 9F*). Alongside, we quantified by immunofluorescence the number of cells that accumulate PC VII and I intracellularly. For this, we counted at least 30 cells in each of five random fields. In STX18 and STX5 knockdown samples more than 80% of the cells accumulated PC VII in the ER compared to 25% of the cells in the control (*Figure 9D,E*). These numbers are comparable to SLY1- or TANGO1-depleted cells. Accumulation of PC I was only detectable in STX5-depleted cells (80%). STX18, like TANGO1 or SLY1 knockdown did not arrest PC I in the ER (*Figure 9G,H*).

## Discussion

Is the export of bulky collagens from the ER mediated by the standard COPII machinery? How is the size of the COPII coats regulated to generate a mega carrier for the export of collagens that can be up to 450 nm of rigid unbendable structures? How are the secretory collagens connected with the cytoplasmic COPII machinery? Are all the bulky collagens exported by a common secretory pathway? Our new data show that knockdown of SAR1 A and B inhibits the export of PC VII from the ER. This,

**Table 1.** SNAREs involved in the early secretory pathway

| Name | t- or v-SNARE | Sub-cellular localization | Secretion pathway | Cargoes associated |
|------|---------------|---------------------------|-------------------|--------------------|
| STX 18 | t | ER (*Nakajima et al., 2004*; *Itakura et al., 2012*) | Golgi to ER, ER to Golgi | Collagen VII (this study), ssGFP (*Gordon et al., 2010*) |
| STX 17 | t | ER (*Itakura et al., 2012*) | Autophagy | GFP-LC3 (*Itakura et al., 2012*) |
| USE1/P31 | t | ER (*Nakajima et al., 2004*; *Okumura et al., 2006*) | Golgi to ER | ERGIC-53, KDEL-R (*Aoki et al., 2009*) |
| SEC20 | t | ER (*Nakajima et al., 2004*) | Golgi to ER, ER to Golgi | ssGFP (*Gordon et al., 2010*) |
| STX 5 | t | Golgi (*Rowe et al., 1998*; *Aoki et al., 2009*) | ER to Golgi, Golgi to ER | Collagen VII (this study), Collagen I (this study), ssGFP (*Gordon et al., 2010*) |
| BET1 | t | ER-ERGIC-Golgi (*Hay et al., 1996*; *Uemura et al., 2009*) | ER to Golgi | Chylomicrons (*Siddiqi et al., 2006*) |
| BET1L | t | Golgi (*Tai et al., 2004*) | unknown | |
| SEC22B | v | ER-ERGIC-Golgi (*Hay et al., 1996*; *Tai et al., 2004*) | ER to Golgi, Golgi to ER | ssGFP (*Gordon et al., 2010*) |
| YKT6 | v | ER-ERGIC-Golgi (*Zhang and Hong, 2001*; *Volchuk et al., 2004*) | ER to Golgi, Golgi to ER | ssGFP (*Gordon et al., 2010*) |
| VTI1a | v | ER-ERGIC-Golgi (*Flowerdew and Burgoyne, 2009*) | ER to Golgi | Chylomicrons (*Siddiqi et al., 2006*) |

t, targeting; v, vesicular; ER, endoplasmic reticulum; ERGIC, ER to Golgi intermediate compartment; ssGFP, signal sequence GFP.

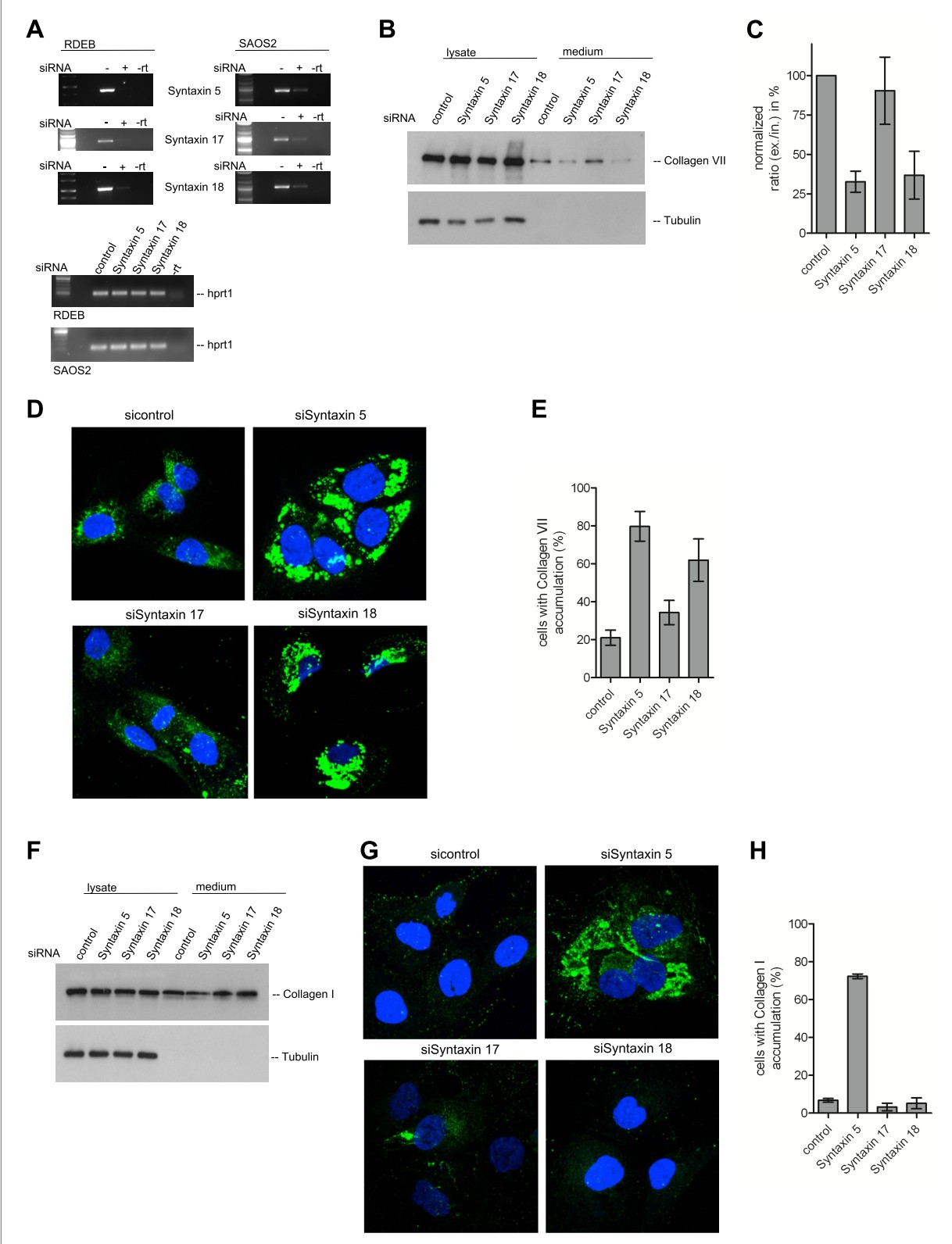

**Figure 9**. The t-SNARE Syntaxin 18 is necessary for Procollagen VII but not Procollagen I export. To evaluate PC VII (**B–E**) and PC I export (**F–H**), RDEB/
FB/C7 and Saos2 cells were transfected with siRNAs directed against STX5, STX17, STX18 or a scrambled siRNA. (**A**) Knockdown efficiency was assessed
48 hr after transfection by RT-PCR. PC VII secretion (**B**) and PC I secretion (**F**) was measured by western blotting cell lysates and supernatants collected
*Figure 9. Continued on next page*

*Figure 9. Continued*
for 20 hr in the presence of ascorbic acid from RDEB/FB/C7 and Saos2 cells, respectively. Equal protein loading and cell lysis was controlled by blotting with an anti-Tubulin antibody. (**C**) Intensities of Collagen VII in the lysate and the supernatant was recorded by densitometry in four independent experiments. The ratio of external vs internal Collagen VII was normalized to quantify secretion in control cells as 100%; Error bars: standard error of the mean (SEM). (**D** and **G**) siRNA treated RDEB/FB/C7 cells were seeded on coverslips and 20 hr after addition of ascorbic acid, cells were fixed and visualized with Collagen VII (**D**) or Collagen I (**G**) antibodies (green), and DAPI (blue) by fluorescence microscopy. The percentage of cells that accumulate PC VII (**E**) or PC I (**H**) intracellularly was determined by counting at least 30 cells in five random fields. The number of cells accumulating PC VII in the case of STX5 and STX18 siRNA was significantly different from control cells (p<0.05). Accumulation of PC VII in STX17 siRNA cells was not significantly different from the control situation (p>0.1). Error bars: standard error of the mean (SEM).

combined with the data on the involvement of SEC31 ubiquitination for the export of PC I from the ER (*Jin et al., 2012*) and Sedlin, a component of the TRAPP II complex that affects the GTPase activity of SAR1 and promotes PC I and II export (*Venditti et al., 2012*), strongly indicates that SAR1 dependent recruitment of COPII components is required for general collagen export from the ER.

## Secretory cargo sorting in the ER

A large number of secretory cargoes bind receptors of the ERGIC53, ERV, and p24 families; the receptors in turn bind the inner COPII coat components and are then packed into a standard COPII vesicle (*Zanetti et al., 2012*). Our previous findings indicate that TANGO1, which localizes to ER exit sites, binds PC VII in the lumen and SEC23/24 of the COPII coats on the cytoplasmic site (*Saito et al., 2009*). The function of TANGO1 is facilitated by cTAGE5 (*Saito et al., 2011*). cTAGE5 and TANGO1 interact through their second (proximal to the proline rich domain) coiled–coiled domains at the ER exit sites. cTAGE5 does not have a luminal domain and cannot therefore bind PC VII directly (*Saito et al., 2011*). The TANGO1–cTAGE5 complex is proposed to stall the COPII vesicles at the ER exit sites as they are being packed with PC VII and this procedure is proposed to result in the production of a transport intermediate for the export of PC VII from the ER (*Malhotra and Erlmann, 2011*). The process by which PC I in the lumen of the ER is connected to COPII coat components remains unclear.

In our experimental conditions, SLY-1 and the ER-associated t-SNARE STX18 are specifically required for the export of PC VII but not of the general cargoes and PC I. This strongly indicates the existence of different export pathways from the ER (*Figure 10A*). These findings also highlight that the sorting of cargoes into different export routes is not driven solely by the size of the secretory cargoes. While the mechanism regulating this sorting event remains unclear it is worth highlighting that TANGO1 in *Drosophila* is localized to the basal endoplasmic reticulum specific exit sites (*Lerner et al., 2013*). Moreover, TANGO1 in *Drosophila* is required for the export of Collagen IV, which is secreted into the basal face of the cells (*Lerner et al., 2013*). It is therefore tempting to propose that secretory cargoes are sorted in the lumen of the ER based on their final destination.

Human cells contain a second SLY1 like protein (SCFD2, Gene ID: 152579). A TANGO1 like protein called MIA2 that is expressed only in the liver and small intestine has also been identified (*Pitman et al., 2011*). The cargoes exported by these proteins are not known but they could, in combination with SAR1, facilitate the export of other collagens, including Collagen I and perhaps many other bulky molecules, from the ER.

At this juncture, it is also important to comment on our original finding that TANGO1 inhibited the secretion of an artificial cargo ss-HRP in *Drosophila* S2 cells (*Bard et al., 2006*). There is no evidence that TANGO1 binds HRP in the lumen of the ER and yet both knockdown of TANGO1 and SLY1, as shown here, affect ss-HRP secretion in mammalian cells. We suggest that the exogenously expressed ss-HRP or ss-GFP exits the ER by any route available and blocking any one, for example by TANGO1 or SLY1 knockdown, affects its export because of a block in one of the possible exit routes.

## The requirement of an ER attached t-SNARE in cargo export from the ER

Our surprising finding is that SLY1, which is known to interact with STX18, a t-SNARE for membrane fusion (*Hatsuzawa et al., 2000*), is required for the export of PC VII from the ER. The function of SLY1, which is a member of the STXBP/unc-18/SEC1 protein family, remains unclear. For example, it is not known whether SLY1 regulates the activity of the SNARE complex, or helps with the fusion or the disassembly of the SNAREs (*Carr and Rizo, 2010*). SLY1 at the ER -Golgi interface has been shown

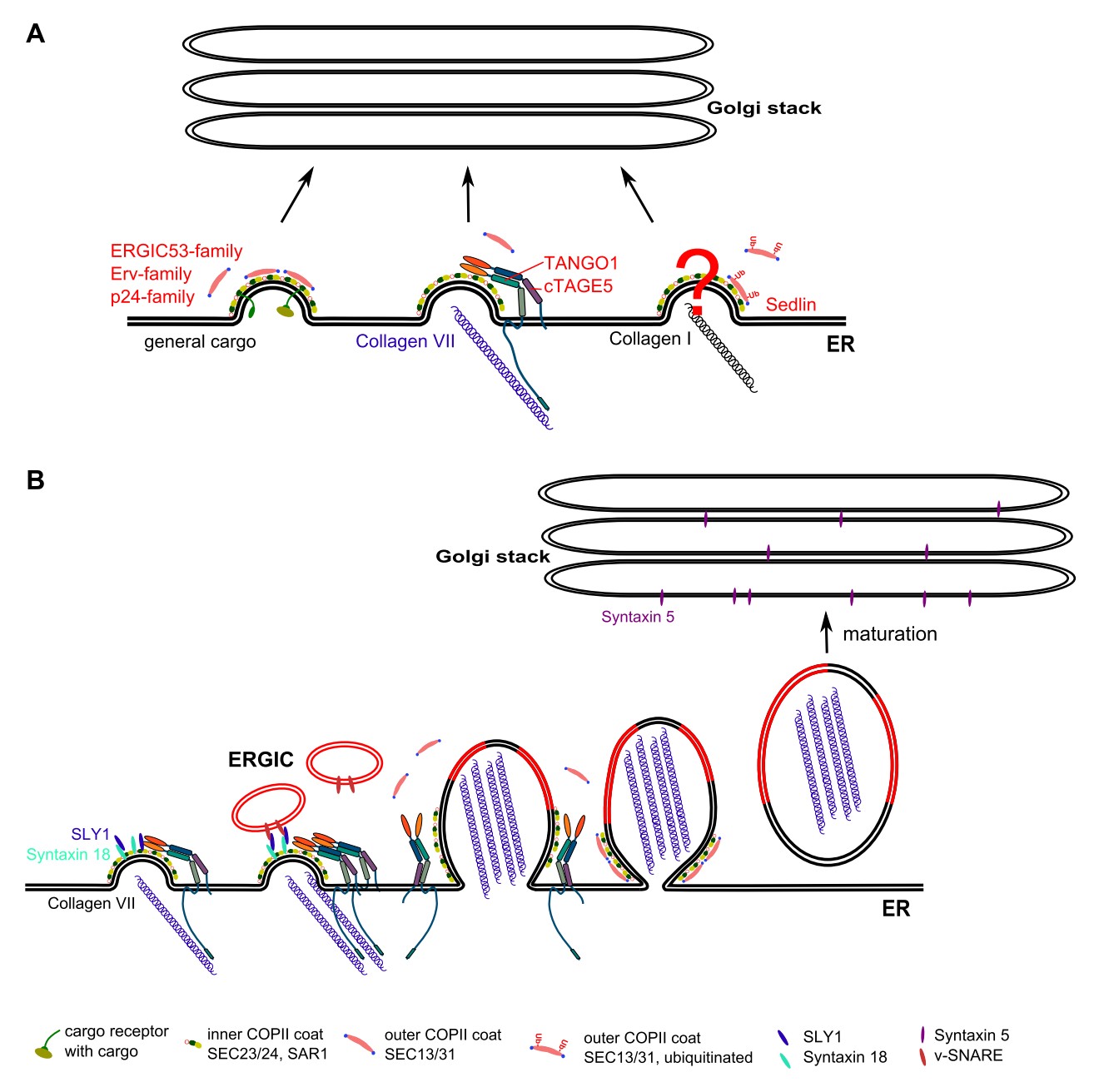

**Figure 10**. A working model for procollagen export from the ER. (**A**) Multiple exit routes from the ER. Cargo receptors of the ERGIC53, ERV, and p24 families bind secretory cargoes in the lumen and the inner COPII coat on the cytoplasmic side of the ER. These receptors bound to cargoes are collected into COPII vesicles for export from the ER. TANGO1 connects PC VII in the lumen of the ER with COPII coats on the cytoplasmic side of the ER. The mechanism by which PC I is connected with the COPII components is not known. Sedlin and ubiquitination of SEC31 are required for PC I secretion but their role in PC VII export is not known. (**B**) Building a Procollagen VII containing mega carrier by fusion of recycling ERGIC membranes. Post concentration of PC VII by TANGO1 at the ER exit sites, many of the ER exit sites are concentrated to generate a patch enriched in PC VII. This patch containing STX18 then promotes SLY1 dependent fusion of membranes from the ERGIC. Accretion of membranes by this process grows PC VII-enriched ER patch. The COPII coats and the TANGO1 remain at the neck of the growing patch. These components at the neck promote fission and the resulting mega container enriched in PC VII and ERGIC membrane components is in fact the first post ER compartment that matures to move PC VII forward.

to interact with a large number of SNAREs and could thus regulate a number of different SNARE complexes (*Dascher and Balch, 1996*; *Steegmaier et al., 2000*; *Yamaguchi et al., 2002*). The t- and v-SNAREs of the ER Golgi anterograde and retrograde trafficking routes are summarized in *Table 1*.

So far the SNARE complexes reported for the forward trafficking from ER to Golgi are complexes of Golgi t-SNAREs and a specific v-SNARE. However, there is no evidence that the ER attached t-SNAREs such as STX17, STX18, Sec20 and USE1/P31 are required for the forward transport to the Golgi complex. It could be argued that SLY1 and STX18 function to recycle a specific component, a v-SNARE; for example, from ERGIC/early Golgi cisternae and without which, the cells terminate the export of PC VII out of the ER. However, cycling of KDEL containing protein BIP and ERGIC53 as well as the export of other secretory cargoes including the bulky PC I is unaffected under these experimental conditions. Additionally, we do not see any obvious accumulation of PC VII containing large vesicles in between the ER and the Golgi. We therefore favor the possibility that SLY1 and STX18 are required for the fusion of membranes to the ER to generate an export carrier for PC VII. The working hypothesis for the export of PC VII from the ER follows (*Figure 10B*).

1. A SAR1 dependent reaction initiates the assembly of COPII subunits SEC23/24 at the ER exit sites.
2. Recruitment of TANGO1-cTAGE5 complex into this reaction with its binding of PC VII in the lumen and SEC23/24 on the cytoplasmic site.
3. The binding of TANGO1-cTAGE5 to SEC23/24 retards the recruitment of SEC13/31 and prevents premature pinching of membranes (*Saito et al., 2009*; *Malhotra and Erlmann, 2011*; *Saito et al., 2011*). These ER exit sites containing TANGO1 bound PC VII are then concentrated to generate a specific patch, which for the sake of simplicity we call PCP (PC VII concentrated patch).
4. A special pool of recycling membranes; for example, the ERGIC, then fuse with PCP by a reaction that requires SLY1 and STX18.
5. As the PCP grows, TANGO1 dissociates from the collagen, which then promotes the recruitment of SEC13/31 to the neck. This reaction also prevents the packing of TANGO1-cTAGE 5 into the PC VII enriched bud (PEB). The neck of this large PEB, concentrated in COPII coats, undergoes fission by the same mechanism as that of a standard COPII vesicle.

The PC VII enriched large membrane compartment separated from the ER also contains recycled ERGIC membranes. This membrane bound compartment in principle is the first post ER compartment that matures to move PC VII in the anterograde direction for secretion.

Interestingly, Luini et al. have reported that export of PC I from the ER requires its concentration at a domain close to the ER exit sites, which they suggest protrudes and then separates from the ER to generate a PC I enriched carrier (*Mironov et al., 2003*). Their model is reminiscent of our scheme in which we place molecular components such as TANGO1 for the concentration of PC VII followed by the fusion of membranes to grow the PC VII enriched ER domain. However, the TANGO1 mediated concentration of PC VII requires the involvement of COPII components, SEC23/24. Additionally, the protrusion visualized by the microscopy based analysis might correspond to growth of ER patch by SLY1 and STX18 dependent fusion of the recycling (ERGIC) membranes. The challenge is to find proteins that connect PC I with the COPII machinery, the SNAREs required for the trafficking of PC I, and the visualization of membranes that we propose are added to grow the patch of PC VII containing ER membrane into a vectorial carrier en route to the cell surface for secretion.

## Materials and methods

### cDNA cloning and constructs

TANGO1 constructs were as described previously (*Saito et al., 2009*).

Full-length *SLY1* was amplified by RT-PCR from the total mRNA of HeLa cells. *SLY1* was cloned into the retroviral vector pLJM1-GFP (Addgene, Cambridge MA) using Gibson cloning strategy (*Gibson et al., 2009*). SnapGene software (from GSL Biotech, Chicago, IL; available at www.snapgene.com) was used for molecular cloning procedures. The Ii-FRAP-HA plasmid has been previously described (*Pecot and Malhotra, 2006*). The FKBP-ERGIC53-GFP plasmid was a gift from Dr Hauri (*Ben-Tekaya et al., 2005*).

### Cell culture

RDEB/FB/C7 cells, HeLa, Het1a, and Saos2 cells were grown at 37°C with 5% $CO_2$ in complete DMEM with 10% fetal bovine serum. Plasmids were transfected in HeLa, with TransIT-HeLaMONSTER (Mirus Bio LLC, Madison, WI) or X-tremeGENE9 (Roche, Indianapolis, IN) according to the manufacture's protocols. siRNAs were transfected in HeLa, RDEB/FB/C7, Het1a and Saos2 with Hiperfect (Qiagen, Venlo, Netherlands) according to the manufacture's protocols. In the case of RDEB/FB/C7, siRNAs

were transfected twice, on day 0 and on day 1. For lentiviral infection of *SLY1-GFP* into RDEB/FB/C7 cells, lentiviral particles were produced by co-tranfecting HEK293 cells with a third generation packaging vector pool and pJLM1-SLY1-GFP using TransIT-293 (Mirus Bio LLC). At 72 hr post transfection the viral supernatant was harvested, filtered, and directly added to RDEB/FB/C7 cells.

## siRNA oligos

siRNAs for SLY1, TANGO1, STX5, STX17, STX18, and SAR1 A and B were purchased from Eurofins MWG Operon (Huntsville, AL): SLY1 siRNA- AGACUUAUUGAUCUCCAUA; TANGO1 siRNA- GAUAAG GUCUUCCGUGCUU; STX5 siRNA- GGACAUCAAUAGCCUCAAC; STX17 siRNA- GACUGUUGG UGGAGCAUUU; STX18 siRNA- CAGGACCGCUGUUUUGGAUUU; for knockdown of SAR1A+B a pool of 4 siRNAs were used as previously described (*Cutrona et al., 2013*). Control siRNAs consisted of a pool of ON-TARGETplus Non-Targeting siRNAs (D-001810-10-05, Thermo Scientific, Waltham, MA).

## Antibodies

Antibodies used in conventional western blotting, immunofluorescence, and immunoelectron microscopy are the following: SLY1, Collagen VII, Collagen I, GRP78/BIP, SEC23A (Abcam, Cambridge, UK); TANGO1 (LifeSpan Biosciences, Seattle, WA); SAR1 (Milipore, Billerica, MA); HSP47 (Enzo Life Sciences, Farmingdale, NY); SEC31, GM130 (BD Biosciences, San Jose, CA); β-COP, myc, Tubulin, Actin (Sigma-Aldrich, St. Louis, MO); HA (Covance, Princeton, NJ); ε-COP (a gift from Dr Rothman).

## Immunoprecipitation

HeLa cells were transfected with a plasmid encoding MycHis-tagged cytoplasmatic tail of TANGO1. 24 hr post transfection proteins were cross linked by incubation with the cell permeable cross linker DSP (Pierce, Rockford, IL USA) for 30 min at room temperature. Residual cross linker was quenched by adding Tris-HCl to a final concentration of 150 mM. The cells were scraped into lysis buffer (50 mM Tris, pH 7.5, 150 mM NaCl, 1% TX100, 1 mM sodium orthovanadate, 10 mM sodium fluoride, and 20 mM β-glycerophosphate, 10 mM Imidazol) and lysates were clarified by centrifugation at 16,000×$g$ for 10 min at 4°C. The lysates were then incubated with Ni$^+$-Agarose (Qiagen) beads for 2 hr at 4°C on a rotating platform. The beads were washed three times with lysis buffer and the specifically bound proteins eluted using PBS containing 500 mM Imidazol.

## Collagen secretion

RDEB/FB/C7 or Saos2 cells were transfected with siRNA as described above. 24 hr after the last siRNA transfection, the medium was replaced with a fresh medium containing 2 μg/ml ascorbic acid. Fresh medium containing ascorbic acid was added again to the cells 20 hr before being collected. Upon collection, the media were centrifuged at low speed to remove any cell in suspension and the supernatant boiled for 5 min with Laemmli SDS-sample buffer. For cell lysis, the cells were washed with PBS, lysed and centrifuged at 14,000 rpm for 15 min at 4°C. The supernatants were boiled for 5 min with Laemmli SDS-sample buffer. Both media and cell lysate were subjected to SDS-PAGE (6% acrylamide) and western blotting with Collagen VII, Collagen I, and Tubulin antibodies. ImageJ (NIH, Bethesda, Maryland) was used for quantitation.

## Immunofluorescence microscopy

Cells grown on coverslips were fixed either with cold methanol for 10 min at −20°C or with 4% paraformaldehyde in PBS for 10 min followed by permeabilization with 0.2% Triton X100 at room temperature, and then incubated with blocking reagent (Roche) for 30 min at room temperature. Primary antibodies were diluted in blocking reagent and incubated overnight at 4°C. Secondary antibodies conjugated with Alexa 488, Alexa 594 or Alexa 647 were diluted in blocking reagent and incubated for 1 hr at room temperature. For SLY1 localization studies, RDEB/FB/C7 expressing SLY1-GFP were transfected with the indicated siRNAs; after 72 hr after transfection cells were processed for immunofluorescence. Briefly, cells were washed twice with room temperature KHM buffer (125 mM potassium acetate, 25 mM HEPES [pH 7.2], and 2.5 mM magnesium acetate). Cells were then permeabilized by incubation in KHM containing 0.1% Saponin for 5 min on ice followed by wash for 7 min at room temperature with KHM buffer. Cells were subsequently fixed in 4% paraformaldehyde and processed for immunofluorescence microscopy. Images were taken with a Leica TCS SPE confocal with a 63x objective. SEC31 and TANGO1 localization images on *Figure 8* were taken with a Leica TCS SP5II CW-STED system in STED mode with a 100 × 1.4NA objective at a pixel-size of 20 nm using a 592 nm depletion laser and HyD detectors.

## Immunoelectron microscopy

Cryoimmunoelectron microscopy was performed as described previously (*Martinez-Alonso et al., 2005*). Control and SLY1 or TANGO1 depleted RDEB/FB/C7 were fixed with 2% paraformaldehyde and 0.2% glutaraldehyde in 0.1 M sodium phosphate buffer, pH 7.4. After washing in buffer, the cells were pelleted by centrifugation, embedded in 10% gelatin, cooled on ice and cut into 1 mm$^3$ blocks. The blocks were infused with 2.3 M sucrose at 4°C overnight, frozen in liquid nitrogen and stored until cryo-ultramicrotomy. Sections (~50 nm-thick) were cut at −120°C with a diamond knife in a Leica Ultracut T/FCS. Ultrathin sections were picked up in a mix of 1.8% methylcellulose and 2.3 M sucrose (1:1). tc\l 1 'Preparation of immunogold labeled cryosections'. Cryosections were collected on carbon and formvar-coated copper grids and incubated with rabbit polyclonal antibodies against Collagen VII, followed by protein A-gold. After labeling, the sections were treated with 1% glutaraldehyde, counterstained with uranyl acetate pH 7 and embedded in methyl cellulose–uranyl acetate pH 4 (9:1).

## Retrograde transport

HeLa cells were transfected with the respective siRNAs as mentioned above. Past 48 hr, the cells were transfected with FKBP-ERGIC53-GFP and Ii-FRAP-HA plasmids as described previously (*Pecot and Malhotra, 2006*). 24 hr after plasmid transfection, cells were incubated with media only (control), or media + CHX (100 µg/ml) + Rapamycin (200 nM). At the indicated time points, the cells were fixed and analyzed by fluorescence microscopy. ERGIC53 was visualized by GFP fluorescence, and Ii with a HA-antibody and an Alexa594-labeled secondary antibody. Colocalization of the two proteins was assessed in at least 30 cells from five different focus fields of an representative experiment by calculating the Pearson's colocalization coefficient using ImageJ software and the Colocalization Analysis plug-in. Pearson's colocalization coefficient of 1 means total colocalization whereas 0 indicates no colocalization.

## Metabolic labeling

Control and siRNA-treated cells were cultured in DMEM without L-methionine and L-cysteine for 60 min and pulsed with 100 µCi $^{35}$Smethionine (Hartmann Analytics) for 20 min. The cells were washed three times with PBS and chased with DMEM containing 10 mM unlabeled methionine for 2 hr. For BFA treatment, 10 µg/ml BFA was added to the medium for the last 10 min of incubation and kept for the whole experiment. Cells were lysed with PBS containing 1% Triton X-100. 20 µl of cell extracts was mixed with scintillation cocktail, and the radioactivity was determined for normalization. The collected medium was precipitated with TCA and analyzed by SDS-PAGE/autoradiography.

## HRP secretion assay in HeLa cells

HeLa cells stably expressing ss-HRP were cultured in a 12-well plate and transfected with control siRNA, SLY1 or TANGO1-specific siRNA oligos. 72 hr later, cells were washed with medium and incubated with 1 ml of complete medium at 15°C for 2 hr. Then, cells were incubated with 500 µl of complete medium at 37°C in presence of 100 µM CHX. At the time points indicated, the medium and cells were harvested for HRP secretion assay. 50 µl of the medium was collected and cells were lysed in 100 µl of lysis buffer (50 mM Tris, pH 7.4, 150 mM NaCl, 0.1% sodium dodecyl sulfate (SDS), 1% Nonidet P-40 (NP- 40) and 0.5% sodium deoxycholate) supplemented with protease inhibitors, 1 mM Na3VO4, and 25 mM sodium fluoride and centrifuged at 16,000×*g* for 15 min. For the chemiluminescence assay, 50 µl of medium and cell lysate were mixed with an ECL reagent (Thermo Scientific) and luminescence measured with a Victor 3 plate reader (PerkinElmer, Waltham, MA).

## Acknowledgements

We thank the entire Malhotra Lab for help with figures and corrections. We would like to acknowledge the CRG's mass spectrometry facility for sample analysis and the CRG advanced light microscopy facility for help with conventional and STED fluorescent microscopy. P Erlmann was partially funded by a DFG fellowship (ER 681/1-1). V Malhotra is an Institució Catalana de Recerca i Estudis Avançats (ICREA) professor at the Center for Genomic Regulation and the work in his laboratory is funded by grants from Plan Nacional (BFU2008-00414), Consolider (CSD2009-00016), Agència de Gestió d'Ajuts Universitaris i de Recerca (AGAUR) Grups de Recerca Emergents (SGR2009-1488; AGAUR-Catalan Government), and European Research Council (268692).

# Additional information

## Competing interests

VM: Reviewing editor, *eLife*. The other authors declare that no competing interests exist.

## Funding

| Funder | Grant reference number | Author |
| --- | --- | --- |
| Deutsche Forschungsgemeinschaft | DFG fellowship ER 681/1-1 | Patrik Erlmann |
| Institució Catalana de Recerca i Estudis Avançats | | Vivek Malhotra |
| Plan Nacional | BFU2008-00414 | Vivek Malhotra |
| Consolider | CSD2009-00016 | Vivek Malhotra |
| Institució Catalana de Recerca i Estudis Avançats | SGR2009-1488 | Vivek Malhotra |
| European Research Council | 268692 | Vivek Malhotra |

The funders had no role in study design, data collection and interpretation, or the decision to submit the work for publication.

## Author contributions

CN, PE, Conception and design, Acquisition of data, Analysis and interpretation of data, Drafting or revising the article; JV, AJMS, Acquisition of data, Analysis and interpretation of data, Drafting or revising the article; EM-A, JÁM-M, Acquisition of data, Analysis and interpretation of data; VM, Conception and design, Analysis and interpretation of data, Drafting or revising the article

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
