## [Decision Letter]

Thank you for sending your work entitled “SLY1 and Syntaxin 18 mediated membrane fusion is required for Procollagen VII export from the endoplasmic reticulum” for consideration at *eLife.* Your article has been favorably evaluated by a Senior editor and 2 reviewers, one of whom, Suzanne Pfeffer, is a member of our Board of Reviewing Editors.

We discussed the comments before we reached this decision, and we provide the following comments to help you prepare a revised submission.

In this interesting paper, Malhotra and colleagues show that Sly1 interacts with TANGO 1 and is required for Collagen VII ER export, together with Sar1A/B and Syntaxins 5 and 18. Like loss of TANGO1, cells depleted of Sly1 are normal for secretion of general proteins and for collagen I. Loss of Sly1 does not interfere with TANGO1 interaction with cargo on membranes, suggesting (but not proving) that it acts after TANGO1. Syntaxin 5, but not 18, is required for collagen 1 secretion.

Is this suitable for *eLife*? Less is known than one would expect about the molecular basis of ER export in mammalian cells, and this work again highlights the interesting fact that multiple export pathways do indeed exist. If the paper were to teach us what Sly1 does in this process, there would be no hesitation. Nevertheless, the fundamental importance of the process and the molecular clarification of ER export sorting lean in favor of publication by *eLife*.

Issues that need to be addressed:

1) The data are convincing regarding the phenotypes, but the possibility remains that this could be an indirect effect, despite the authors attempt to demonstrate otherwise. Certain cargos, like endogenous PC VII, and overexpressed cargos like ssHRP, do clearly have requirements for additional factors to aid in their incorporation into COPII vesicles. Indeed, TANGO1 is one of these factors. But given the established roles of SLY1, STX18, and other related proteins in mediating vesicle fusion, it has not yet been shown that SLY1 is directly involved in this process. An alternative explanation is that PCVII is an especially sensitive cargo, and its incorporation into COPII vesicles is hindered by general trafficking perturbations that do not affect most other cargos. Furthermore, how do the authors know that membranes are the limiting component for PCVII vesicle formation upon SLY1 knockdown? Might the recycling ERGIC membranes contain specific protein factors that might be important for this process? These are very tricky issues to address, but we offer a suggestion for how the authors might make their case stronger:

Although the authors identify SLY1 as an interactor of TANGO1, this interaction is not explored functionally. Can the authors show that the interaction is important for PCVII to exit the ER? (i.e., identify a mutant in either TANGO1 or SLY1 that disrupts the interaction, and test whether this mutant is functional for PCVII ER export).

2) The authors conclude with a speculative model suggesting that Sly1 helps fuse ERGIC derived membranes to build collagen VII carriers. Equally possible is a model in which the SNAREs drive fusion of the carriers with the target membranes and cannot form without packaging of the SNAREs. This should be discussed as well. Moreover, the authors should summarize what is known about all ER localized SNARE proteins, perhaps in a table, to highlight gaps in our understanding. The authors find that the SM protein SLY1 interacts with TANGO1, and find that SLY1 and STX18 are important for ER export of the large cargo PC VII. The key experimental results in this paper are that SLY1 knockdown dramatically affects the ER exit of PCVII and overexpressed ssHRP, but not ER exit of other cargos like PCI.

3) If SLY1 is directly involved in PCVII ER export, one might expect that levels of SLY1 would be highest in those tissues or cells responsible for secreting PCVII. Is there any evidence in the literature (transcriptomics, proteomics) that SLY1 is particularly abundant in such tissues?

4) One reviewer took issue with the authors' claim that SLY1 and STX18 act at a “post-COPII assembly step”. The evidence for this claim is based on the fact that SEC31 still forms punctae on the same ER compartments where PCVII is accumulating in SLY1-knockdown cells (Figure 8). However, there is no reason to think that these SEC31 punctae represent stalled PCVII vesicle buds. These SEC31 punctae might be unaffected ERES through which other cargos are continuing to exit normally (because the authors show that other cargos are not affected by SLY1-knockdown). SLY1-knockdown may have no affect at all on the observed SEC31 punctae, and may have actually prevented the formation of PCVII-nucleated SEC31 punctae (which would remain unseen). If SLY1 acts downstream of COPII assembly, one would expect to see a proliferation of SEC31 punctae in the SLY1-knockdown cells, corresponding to stalled PCVII buds. Please modify the text accordingly.

---

## [Author Response]

We are happy to receive the supporting comments on our findings describing the export of Procollagen VII from the Endoplasmic Reticulum (ER). The reviewers have requested that we show more on the interaction between TANGO1 and SLY1. Unfortunately, at present, this is beyond the scope of the paper. TANGO1 coprecipitates with SLY1 only in cells that are chemically crosslinked and we believe they are in a complex of a large number of other proteins. It is, therefore, unclear whether these two proteins bind directly. We were fortunate that we could identify this interaction but defining the mechanism of their binding is not possible at present. We have rewritten the paper extensively to address the reviewers’ concerns and better describe the data rather than a potential model based on our findings. Our data highlights the existing of separate pathways for the export of Procollagen VII and Procollagen I from the ER. In other words, size alone is not the reason for the separate pathways of procollagen export from the ER. Our revised manuscript highlights this new message. In addition, we have added a table that summarizes our current understanding of the SNARE proteins localized at the ER-Golgi interface. This will help the readers to understand the complexity of this process and appreciate our findings.

*1) The data are convincing regarding the phenotypes, but the possibility remains that this could be an indirect effect, despite the authors attempt to demonstrate otherwise. […] An alternative explanation is that PCVII is an especially sensitive cargo, and its incorporation into COPII vesicles is hindered by general trafficking perturbations that do not affect most other cargos. Furthermore, how do the authors know that membranes are the limiting component for PCVII vesicle formation upon SLY1 knockdown? Might the recycling ERGIC membranes contain specific protein factors that might be important for this process? These are very tricky issues to address*, *but we offer a suggestion for how the authors might make their case stronger:*

*Although the authors identify SLY1 as an interactor of TANGO1, this interaction is not explored functionally. Can the authors show that the interaction is important for PCVII to exit the ER? (i.e. identify a mutant in either TANGO1 or SLY1 that disrupts the interaction, and test whether this mutant is functional for PCVII ER export)*.

The reviewer lists a number of different issues in this point that we address separately.

A) As stated in the response above, it is impossible for us to map the binding interaction between TANGO1 and SLY1. We have changed the Title, Abstract, Introduction, and the Discussion extensively to state more clearly the facts.

B) Effect on specific cargo: a V-SNARE for example. We address this in the discussion.

C) Is PC VII somehow more susceptible to our experimental perturbations? It cannot be because of our data that the export of PC I, which is roughly the same size is not affected by either the loss of TANGO1, SLY1 or STX18. This is now stated more clearly in the text.

*2) The authors conclude with a speculative model suggesting that Sly1 helps fuse ERGIC derived membranes to build collagen VII carriers. Equally possible is a model in which the SNAREs drive fusion of the carriers with the target membranes and cannot form without packaging of the SNAREs. This should be discussed as well. Moreover, the authors should summarize what is known about all ER localized SNARE proteins, perhaps in a table, to highlight gaps in our understanding. The authors find that the SM protein SLY1 interacts with TANGO1, and find that SLY1 and STX18 are important for ER export of the large cargo PC VII. The key experimental results in this paper are that SLY1 knockdown dramatically affects the ER exit of PCVII and overexpressed ssHRP, but not ER exit of other cargos like PCI*.

We state this in the text but also highlight that we favor an alternative model. A table with the known v-and t-SNAREs at the ER-Golgi interface is now included in the paper.

*3) If SLY1 is directly involved in PCVII ER export, one might expect that levels of SLY1 would be highest in those tissues or cells responsible for secreting PCVII. Is there any evidence in the literature (transcriptomics*, *proteomics) that SLY1 is particularly abundant in such tissues?*

No, there is no evidence in the literature that SLY1 or TANGO1 levels are particularly higher in tissues that export PC VII.

*4) One reviewer took issue with the authors' claim that SLY1 and STX18 act at a “post-COPII assembly step”. The evidence for this claim is based on the fact that SEC31 still forms punctae on the same ER compartments where PCVII is accumulating in SLY1-knockdown cells (*Figure 8*). However, there is no reason to think that these SEC31 punctae represent stalled PCVII vesicle buds. These SEC31 punctae might be unaffected ERES through which other cargos are continuing to exit normally (because the authors show that other cargos are not affected by SLY1-knockdown). SLY1-knockdown may have no affect at all on the observed SEC31 punctae, and may have actually prevented the formation of PCVII-nucleated SEC31 punctae (which would remain unseen). If SLY1 acts downstream of COPII assembly, one would expect to see a proliferation of SEC31 punctae in the SLY1-knockdown cells, corresponding to stalled PCVII buds. Please modify the text accordingly*.

We have changed the subheading to “SLY1 knockdown does not affect Sec13/Sec31 localization to the ER exit sites“. It is also important to note that we do not state that these are stalled PC VII buds. All we say is that SLY1 depletion does not affect the connection made by TANGO1 between PC VII and the COPII coats of the exit site. We have deleted our statement that SLY1 acts post COPII coat assembly from the discussion to alleviate the issue raised by the reviewer.